# Prompt-based Logical Semantics Enhancement for Implicit Discourse Relation Recognition

**Chenxu Wang**[1]   **Ping Jian**[*1,2]   **Mu Huang**[1]

[1]School of Computer Science and Technology, Beijing Institute of Technology, Beijing, China
[2]Beijing Engineering Research Center of High Volume Language Information Processing
and Cloud Computing Applications, Beijing Institute of Technology, Beijing, China
{wangchenxu, pjian}@bit.edu.cn, suemh2001@gmail.com

## Abstract

Implicit Discourse Relation Recognition (IDRR), which infers discourse relations without the help of explicit connectives, is still a crucial and challenging task for discourse parsing. Recent works tend to exploit the hierarchical structure information from the annotated senses, which demonstrate enhanced discourse relation representations can be obtained by integrating sense hierarchy. Nevertheless, the performance and robustness for IDRR are significantly constrained by the availability of annotated data. Fortunately, there is a wealth of unannotated utterances with explicit connectives, that can be utilized to acquire enriched discourse relation features. In light of such motivation, we propose a **P**rompt-based **L**ogical **S**emantics **E**nhancement (PLSE) method for IDRR. Essentially, our method seamlessly injects knowledge relevant to discourse relation into pre-trained language models through prompt-based connective prediction. Furthermore, considering the prompt-based connective prediction exhibits local dependencies due to the deficiency of masked language model (MLM) in capturing global semantics, we design a novel self-supervised learning objective based on mutual information maximization to derive enhanced representations of logical semantics for IDRR. Experimental results on PDTB 2.0 and CoNLL16 datasets demonstrate that our method achieves outstanding and consistent performance against the current state-of-the-art models.[1]

## 1  Introduction

Implicit discourse relation recognition (IDRR) focuses on identifying semantic relations between two arguments (sentences or clauses), with the absence of explicit connectives (e.g., *however, be-*

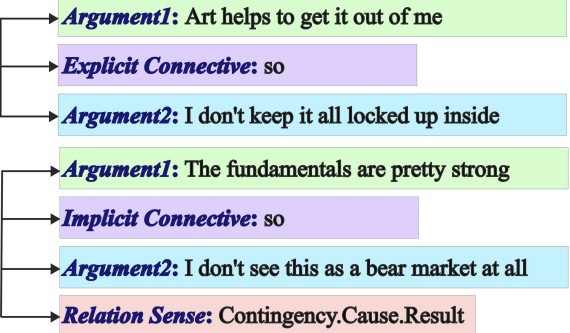

Figure 1: An unannotated instance with an explicit connective and an IDRR instance from Gigaword (Napoles et al., 2012) and PDTB 2.0 (Prasad et al., 2008) respectively. For IDRR, the *Implicit Connective* is absent in the raw text, which are assigned by the annotator based on the semantic relations between *Argument1* and *Argument2*.

*cause*). As a fundamental and crucial task in discourse parsing, IDRR has benefited a multitude of natural language processing (NLP) downstream tasks, such as question answering (Rutherford and Xue, 2015), text summarization (Cohan et al., 2018) and event relation extraction (Tang et al., 2021). Explicit discourse relation recognition (EDRR) has already incontrovertibly demonstrated the remarkable effectiveness by utilizing explicit connectives in discerning discourse relation type (Varia et al., 2019a). However, implicit discourse relation recognition remains a challenging and demanding task for researchers due to the absence of connectives.

Early works based on traditional machine learning are dedicated to the extraction of intricate syntax features (Lin et al., 2009; Park and Cardie, 2012). With the rapid development of deep learning, pre-trained language models (PLMs), such as BERT (Devlin et al., 2019) and RoBERTa (Liu et al., 2019), have remarkably improved the performance of IDRR. Recently, there has been a growing inclination among various work to exploit the multi-level hierarchical information (an instance introduced in Appendix A.1) derived from annotated

---

*Corresponding author.

[1]Our code will be released at https://github.com/lalalamdbf/PLSE_IDRR.

senses (Wu et al., 2022; Jiang et al., 2022; Long and Webber, 2022; Chan et al., 2023b). Nevertheless, such usage exhibits an excessive reliance on the annotated data, thereby restricting the potential of IDRR for further expansion to some degree. Fortunately, there exists a plethora of unannotated utterances containing explicit connectives that can be effectively utilized to enhance discourse relation representations.

As depicted in Figure 1, even in the absence of annotated sense, the explicit data can proficiently convey a Contingency.Cause.Result relation, elucidating a strong correlation between the connective and the discourse relation. In expectation of knowledge transfer from explicit discourse relations to implicit discourse relations, Kishimoto et al. (2020) has undertaken preliminary investigations into leveraging explicit data and connective prediction. This work incorporated an auxiliary pre-training task named explicit connective prediction, which constructed an explicit connective classifier to output connectives according to the representation of the [CLS]. Nonetheless, this method falls short in terms of intuitive connective prediction through the [CLS] and fails to fully harness the valuable knowledge gleaned from the masked language model (MLM) task. Besides, Zhou et al. (2022) proposed a prompt-based connective prediction (PCP) method, but their approach relies on *Prefix-Prompt*, which is not inherently intuitive and introduces additional complexity of inference.

In this paper, we propose a novel **P**rompt-based **L**ogical **S**emantics **E**nhancement (PLSE) method for implicit discourse relation recognition. Specifically, we manually design a *Cloze-Prompt* template for explicit connective prediction (pre-training) and implicit connective prediction (prompt-tuning), which achieves a seamless and intuitive integration between the pre-training and the downstream task. On the other hand, previous work (Fu et al., 2022) has revealed that the contextualized representations learned by MLM lack the ability to capture global semantics. In order to facilitate connective prediction, it is important to have a comprehensive understanding of the text's global logical semantics. Inspired by recent unsupervised representation learning with mutual information (Hjelm et al., 2019; Zhang et al., 2020), we propose a novel self-supervised learning objective that maximizes the mutual information (MI) between the connective-related representation and the global logic-related

semantic representation (detailed in Section 3.2). This learning procedure fosters the connective-related representations to effectively capture global logical semantic information, thereby leading to enhanced representations of logical semantics.

Our work focuses on identifying implicit inter-sentential relations, thus evaluating our model on PDTB 2.0 (Prasad et al., 2008) and CoNLL16 (Xue et al., 2016) datasets. Since the latest PDTB 3.0 (Webber et al., 2019) introduces numerous intra-sentential relations, there is a striking difference in the distribution between inter-sentential and intra-sentential relations reported by (Liang et al., 2020). Given this issue, we don't conduct additional experiments on PDTB 3.0.

The main contributions of this paper are summarized as follows:

- We propose a prompt-based logical semantics enhancement method for IDRR, which sufficiently exploits unannotated utterances with connectives to learn better discourse relation representations.

- Our proposed connective prediction based on *Cloze-Prompt* seamlessly injects knowledge related to discourse relation into PLMs and bridges the gap between the pre-training and the downstream task.

- Our method, aiming at capturing global logical semantics information through MI maximization, effectively results in enhanced discourse relation representations.

## 2 Related Work

**Prompt-based Learning** Along with the booming development of large-scale PLMs like BERT (Devlin et al., 2019), RoBERTa (Liu et al., 2019), T5 (Raffel et al., 2020) and GPT3 (Brown et al., 2020), prompt-based learning has become a new paradigm in the filed of natural language processing (Liu et al., 2021). In contrast to the fine-tuning paradigm, prompt-based methods reformulate the downstream tasks to match the pre-training objective. For example, Schick and Schütze (2021) converted a diverse set of classification problems into cloze tasks by constructing appropriate prompts and verbalizers that associate specific filled words with predicted categories.

**Representation Learning with MI** Methods based on mutual information have a long history in unsupervised representation learning, such as

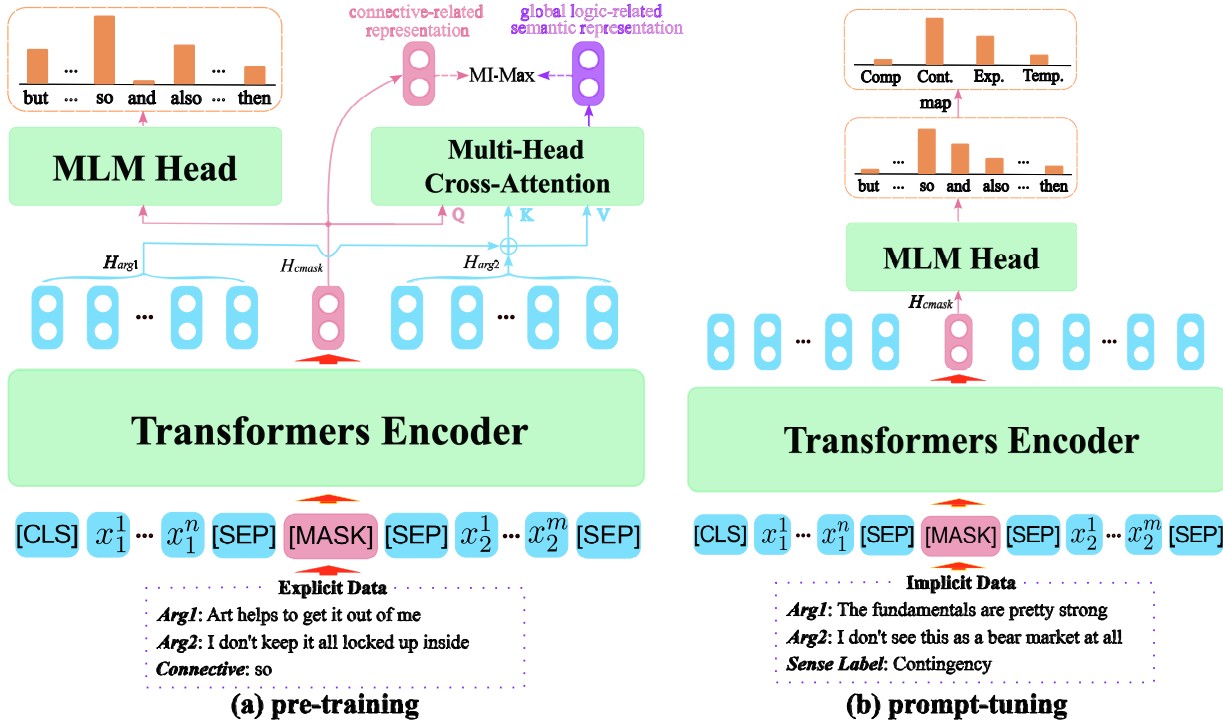

Figure 2: The overall architecture of our framework. It consists of the pre-training phase (a) and the prompt-tuning phase (b). In the pre-training phase, explicit data is automatically extracted from an unannotated corpus, while in the prompt-tuning phase, implicit data is randomly selected from the training data.

the infomax principle (Linsker, 1988; Bell and Sejnowski, 1995) and ICA algorithms (Hyvärinen and Pajunen, 1999). Alemi et al. (2016) was the first to integrate MI-related optimization into deep learning models. From then on, numerous studies have demonstrated the efficacy of MI-maximization for unsupervised representation learning (Hjelm et al., 2019; Zhang et al., 2020; Kong et al., 2019). Inspired by these superior works, we explore how to derive enhanced discourse relation representations by incorporating global logical semantic information through MI-maximization.

**Implicit Discourse Relation Recognition** For this task, many prior works based on deep neural network have proposed various approaches to exploiting better semantic representation of arguments, such as shallow CNN (Zhang et al., 2015) and LSTM with multi-level attention (Liu and Li, 2016). Recently, PLMs have substantially improved the performance of IDRR by leveraging their robust contextualized representations (Shi and Demberg, 2019; Liu et al., 2020). Furthermore, Kishimoto et al. (2020) proposed an auxiliary pre-training task through unannotated explicit data and connective prediction. Zhou et al. (2022) proposed a connective prediction (PCP) method based on *Prefix-Prompt*. Regrettably, their approaches fail to

capitalize on the potential of data and PLMs. On the other hand, the multi-level hierarchical information has been explored by (Wu et al., 2022; Jiang et al., 2022; Long and Webber, 2022; Chan et al., 2023b). However, the efficacy of these methods remains significantly constrained by the availability of annotated data. Therefore, our work is dedicated to the proficient utilization of unannotated explicit data via prompt-based connective prediction and MI-maximization.

## 3 Methodology

Figure 2 shows the overall architecture of our framework.[2] In the following sections, we will elaborate on our methodologies including a *Cloze-Prompt* template based on connective prediction, a robust pre-training approach to incorporating explicit data, and an effective prompt-tuning method by leveraging implicit data.

### 3.1 Prompt Templatize

As Figure 2 illustrates, the input argument pair $x = (Arg1, Arg2)$ is reformulated into a *Cloze-*

---

[2]In the pre-training phase, Figure 2 does not depict the Masked Language Mask (MLM) task for universal words, yet it still remains an integral part of our framework to enhance the universal semantics of discourse relations for PLMs.

*Prompt* template by concatenating two arguments and inserting some PLM-specific tokens such as [MASK], [CLS] and [SEP]. The [MASK] token is mainly used to predict the connective between two arguments, while the [CLS] and [SEP] tokens are inserted to indicate the beginning and ending of the input. In addition, the [SEP] tokens also serve the purpose of delineating the boundaries between two arguments.

## 3.2 Pre-training

In this section, we will provide a comprehensive introduction to the pre-training data and tasks, including Connectives Mask (CM), Masked Language Model (MLM), Global Logical Semantics Learning (GLSL).

**Pre-training Data** Acquiring implicit discourse relational data poses an immensely arduous undertaking. Fortunately, explicit discourse relation data is considerably more accessible owing to the presence of connectives. Therefore, we approximately collected 0.56 million explicit argument pairs from an unannotated corpus renowned as Gigaword (Napoles et al., 2012) by leveraging its vast collection of connectives. The collected data is divided into training and validation sets in a 9:1 ratio.

**Connectives Mask** Based on the strong correlation observed between explicit connectives and discourse relations, we propose the "Connectives Mask" task as a pivotal approach to incorporating discourse logical information into PLMs. This task not only involves learning the intricate representations of connectives but also serves as a vital bridge between the realms of the pre-training and the downstream task, that facilitates seamless knowledge transfer. For the mask strategy, we replace the connective with (1) the [MASK] token 90 % of the time (2) the unchanged token 10% of the time. The cross-entropy loss for Connectives Mask (CM) is as follows:

$$\mathcal{L}_{CM} = \frac{1}{N} \sum_{i=1}^{N} -logP(c_i \,|\, T(x_i)) \quad (1)$$

where $N$ denotes the number of training examples, $T$ embodies a prompt template designed to transform the input argument pair $x$, $c$ represents the connective token between the argument pair, and $P$ estimates the probability of the answer word $c$.

**Masked Language Model** In order to enhance the universal semantics of discourse relations for PLMs, we still retain the Masked Language Model (MLM) task for universal words. The cross-entropy loss for MLM is as follows:

$$\mathcal{L}_{MLM} = \frac{1}{N} \sum_{i=1}^{N} \underset{j \in masked}{mean} -logP(u_{i,j}|T(x_i)) \quad (2)$$

where $N$ and $T$ remain consistent with Equation (1), $u$ denotes the universal token within the argument pair, and $P$ approximates the probability of the answer word $u$.

**Global Logical Semantics Learning** To address the local dependencies of predicted connectives representations and capture global semantic information with logical coherence, we propose a novel method based on MI maximization learning. Give an argument pair $x = (Arg1, Arg2)$ and the *Cloze-Prompt* template $T(\cdot)$, we feed $T(x)$ through a Transformer (Vaswani et al., 2017) encoder to acquire contextualized token representations $H$. Specifically, we design a Multi-Head Cross-Attention (MHCA) module to extract the global logic-related semantic representation. As illustrated in the part (a) of Figure 2, we separate $H$ into $H_{arg1}$, $H_{cmask}$ and $H_{arg2}$, denoting as the contextualized representations of $arg1$, *connective mask* and $arg2$, respectively. The global logic-related representation $H_{Glogic}$ is calculated through MHCA module as follows:

$$Q = H_{cmask} \quad K = V = H_{arg1} \oplus H_{arg2} \quad (3)$$
$$H_{Glogic} = \text{MultiHead}(Q, K, V) \quad (4)$$

The learning objective is to maximize the mutual information between the connective-related representation $H_{cmask}$ and the global logic-related representation $H_{Glogic}$. Due to the notorious intractability of MI estimation in continuous and high-dimensional settings, we typically resort to maximizing lower bound estimators of MI. In our method, we adopt a Jensen-Shannon MI estimator suggested by (Hjelm et al., 2019; Zhang et al., 2020):

$$\begin{aligned}\widehat{\mathcal{I}}_\omega^{JSD}&(H_{Glogic}\,;H_{cmask}) \coloneqq \\ &E_{\mathbb{P}}[-sp(-T_\omega(H_{Glogic}, H_{cmask})] \quad (5)\\ &-E_{\mathbb{P}\times\tilde{\mathbb{P}}}[sp(T_\omega(H'_{Glogic}, H_{cmask})]\end{aligned}$$

where $T_\omega$ is a discrimination function parameterized by a neural network with learnable parameters $\omega$. It takes a pair of the global logic-related

representation $H_{Glogic}$ and the connective-related representation $H_{cmask}$ as input and generates the corresponding score to estimate $\widehat{\mathcal{I}}_\omega^{JSD}$. $H'_{Glogic}$ is a negative example randomly sampled from distribution $\tilde{\mathbb{P}} = \mathbb{P}$, and $\mathrm{sp}(z) = \log(1+e^z)$ is the softplus function. The end-goal training objective is maximizing the MI between $H_{Glogic}$ and $H_{cmask}$, and the loss is formulated as follows:

$$\mathcal{L}_{GLSL} = -\widehat{\mathcal{I}}_\omega^{JSD}(H_{Glogic}; H_{cmask}) \quad (6)$$

Through maximizing $\widehat{\mathcal{I}}_\omega^{JSD}$, The connective-related representation $H_{cmask}$ is strongly encouraged to maintain a high level of MI with its global logic-related representation. This will foster $H_{cmask}$ to incorporate comprehensive logical semantic information, leading to enhanced discourse relation representations.

The overall training goal is the combination of *Connectives Mask* loss, *MaskedLM* loss, and *Global Logical Semantics Learning* loss:

$$\mathcal{L} = \mathcal{L}_{CM} + \mathcal{L}_{MLM} + \mathcal{L}_{GLSL} \quad (7)$$

### 3.3 Prompt-tuning

As depicted in the part (b) of Figure 2, the input argument pair $x = (Arg1, Arg2)$ and the *Cloze-Prompt* template $T(\cdot)$ exhibit a remarkable consistency with the pre-training phase. This noteworthy alignment indicates that the prompt-tuning through connective prediction fosters the model to fully leverage the profound reservoir of discourse relation knowledge acquired during the pre-training phase. In order to streamline the downstream task and avoid introducing additional trainable parameters, the MHCA module isn't used in the prompt-tuning phase. Besides, we construct a verbalizer mapping connectives to implicit discourse relation labels.

**Verbalizer Construction** A discrete answer space, which is a subset of the PLM vocabulary, is defined for IDRR. Given that the implicit discourse relational data has already been annotated with appropriate connectives in PDTB 2.0 (Prasad et al., 2008) and CoNLL16 (Xue et al., 2016) datasets, we manually select a set of high-frequency and low-ambiguity connectives as the answer words of the corresponding discourse relations. Therefore, we construct the verbalizer mapping connectives to implicit discourse relation labels for PDTB 2.0 in Table 1, and the verbalizer for CoNLL16 is shown in Appendix A.3.

| Top-level | Second-level | Connectives |
|---|---|---|
| Comparison | Concession | however
although, though |
| | Contrast | but |
| Contingency | Cause | because, so, thus
consequently, therefore |
| | Pragmatic cause | as, since |
| Expansion | Alternative | instead, rather |
| | Conjunction | and, also
fact, furthermore |
| | Instantiation | instance, example |
| | List | finally |
| | Restatement | specifically
indeed, particular |
| Temporal | Asynchronous | then, after, before |
| | Synchrony | meanwhile, when |

Table 1: The verbalizer mapping connectives between implicit discourse relation labels and connectives on PDTB 2.0 dataset, consisting of four top-level and 11 second-level senses.

The cross-entropy loss for the prompt-tuning is as follows:

$$\mathcal{L} = \frac{1}{N}\sum_{i=1}^{N} -logP(l_i = V(c_i) \,|\, T(x_i)) \quad (8)$$

where $N$ and $T$ remain consistent with Equation (1), $V$ denotes the verbalizer mapping the connective $c$ to implicit discourse relation label $l$, and $P$ estimates the probability of the gold sense label $l$.

## 4 Experiment Settings

### 4.1 Datasets

**The Penn Discourse Treebank 2.0 (PDTB 2.0)** PDTB 2.0 is a large scale corpus containing 2,312 Wall Street Journal (WSJ) articles (Prasad et al., 2008), that is annotated with information relevant to discourse relation through a lexically-grounded approach. This corpus includes three levels of senses (i.e., classes, types, and sub-types). We follow the predecessors (Ji and Eisenstein, 2015) to split sections 2-20, 0-1, and 21-22 as training, validation, and test sets respectively. We evaluate our framework on the four top-level implicit discourse classes and the 11 major second-level implicit discourse types by following prior works (Liu et al., 2019; Jiang et al., 2022; Long and Webber, 2022). The detailed statistics of the top-level and second-level senses are shown in Appendix A.2.

**The CoNLL-2016 Shared Task (CoNLL16)** The CoNLL 2016 shared task (Xue et al., 2016) provides more abundant annotation for shadow discourse parsing. This task consists of two test sets,

the PDTB section 23 (CoNLL-Test) and Wikinews texts (CoNLL-Blind), both annotated following the PDTB annotation guidelines. On the other hand, CoNLL16 merges several labels to annotate new senses. For instance, *Contingency.Pragmatic cause* is merged into *Contingency.Cause.Reason* to remove the former type with very few samples. There is a comprehensive list of 14 sense classes to be classified, detailed cross-level senses as shown in Appendix A.3.[3]

## 4.2 Implementation Details

In our experiments, we utilize the pre-trained RoBERTa-base (Liu et al., 2019) as our Transformers encoder and employ AdamW (Loshchilov and Hutter, 2017) as the optimizer. The maximum input sequence length is set to 256. All experiments are implemented in PyTorch framework by HuggingFace transformers[4] (Wolf et al., 2020) on one 48GB NVIDIA A6000 GPU. Next, we will provide details of the pre-training and prompt-tuning settings, respectively.

**Pre-training Details**   During the pre-training phase, we additionally add a MHCA module and a discriminator $T_\omega$ to facilitate MI maximization learning. The MHCA module is a multi-head attention network, where the number of heads is 6. The discriminator $T_\omega$ consists of three feedforward layers with ReLU activation, and the detailed architecture is depicted in Appendix A.4. We pre-train the model for two epochs, and other hyper-parameter settings include a batch size of 64, a warmup ratio of 0.1, and a learning rate of 1e-5.

**Prompt-tuning Details**   The implementation of our code refers to OpenPrompt (Ding et al., 2022). We use Macro-F1 score and Accuracy as evaluation metric. The training is conducted with 10 epochs, which selects the model that yields the best performance on the validation set. For top-level senses on PDTB 2.0, we employ a batch size of 64 and a learning rate of 1e-5. For second-level senses on PDTB 2.0 and cross-level senses on CoNLL16, we conducted a meticulous hyper-parameter search due to the limited number of certain senses. The bath size and learning rate are set to 16 and 1e-6, respectively.

---

[3]Some instances in PDTB 2.0 and CoNLL16 datasets are annotated with multiple senses. Following previous works, we treat them as separate examples during training. At test time, a prediction is considered correct when it aligns with one of the ground-truth labels.

[4]https://github.com/huggingface/transformers

## 4.3 Baselines

To validate the effectiveness of our method, we compare it with the most advanced baselines currently available. Here we mainly introduce the **Baselines** that have emerged within the past two years.

- **CG-T5** (Jiang et al., 2021): a joint model that recognizes the relation label and generates the target sentence containing the meaning of relations simultaneously.

- **LDSGM** (Wu et al., 2022): a label dependence-aware sequence generation model, which exploits the multi-level dependence between hierarchically structured labels.

- **PCP** (Zhou et al., 2022): a connective prediction method by applying a *Prefix-Prompt* template.

- **GOLF** (Jiang et al., 2022): a contrastive learning framework by incorporating the information from global and local hierarchies.

- **DiscoPrompt** (Chan et al., 2023b): a path prediction method that leverages the hierarchical structural information and prior knowledge of connectives.

- **ContrastiveIDRR** (Long and Webber, 2022): a contrastive learning method by incorporating the sense hierarchy and utilizing it to select the negative examples.

- **ChatGPT** (Chan et al., 2023a): a method based on ChatGPT through the utilization of an in-context learning prompt template.

## 5 Experimental Results

### 5.1 Main Results

Table 2 shows the primary experimental results of our method and other baselines on PDTB 2.0 and CoNLL16 datasets, in terms of Macro-F1 score and Accuracy. Classification performance on PDTB 2.0 in terms of F1 score for the four top-level classes and 11 second-level types is shown in Table 3 and Table 4. Based on these results, we derive the following conclusions:

**Firstly**, our PLSE method has achieved new state-of-the-art performance on PDTB 2.0 and CoNLL16 datasets. Specifically, our method exhibits a notable advancement of 1.8% top-level F1, 3.35% top-level accuracy, 0.45% second-level F1,

| Model | PDTB-Top | | PDTB-Second | | CoNLL-Test | | CoNLL-Blind | |
|---|---|---|---|---|---|---|---|---|
| | *F1* | *Acc.* | *F1* | *Acc.* | *F1* | *Acc.* | *F1* | *Acc.* |
| MANN (Lan et al., 2017) | 47.80 | 57.39 | - | - | - | 40.91 | - | 40.12 |
| RWP-CNN (Varia et al., 2019b) | 50.20 | 59.13 | - | - | - | 39.39 | - | 39.36 |
| BERT-FT (Kishimoto et al., 2020) | 58.48 | 65.26 | - | 54.32 | - | - | - | - |
| BMGF-RoBERTa (Liu et al., 2020) | 63.39 | 69.06 | *35.25* | 58.13 | *40.68* | 57.26 | *28.98* | 55.19 |
| CG-T5 (Jiang et al., 2021) | 57.18 | 65.54 | 37.76 | 53.13 | - | - | - | - |
| LDSGM (Wu et al., 2022) | 63.73 | 71.18 | 40.49 | 60.33 | - | - | - | - |
| PCP (Zhou et al., 2022) | 64.95 | 70.84 | 41.55 | 60.54 | *33.27* | *55.48* | *26.00* | *50.99* |
| GOLF (Jiang et al., 2022) | 65.76 | 72.52 | 41.74 | 61.16 | - | - | - | - |
| DiscoPrompt (Chan et al., 2023b) | 65.79 | 71.70 | 43.68 | 61.02 | 45.99 | 60.84 | **39.27** | 57.88 |
| ContrastiveIDRR (Long and Webber, 2022) | 69.60 | 72.18 | 49.66 | 61.69 | - | - | - | - |
| ChatGPT (Chan et al., 2023a) | 36.11 | 44.18 | 16.20 | 24.54 | - | - | - | - |
| PLSE | **71.40** | **75.43** | **50.11** | **64.00** | **54.69** | **61.49** | 39.19 | **58.35** |

Table 2: The Macro-F1 score (%) and Accuracy (%) are evaluated on PDTB 2.0 and CoNLL16 datasets. Italics numbers indicate the reproduced results from (Chan et al., 2023b). Bold numbers correspond to the best results. Underlined numbers correspond to the second best.

| Model | Comp. | Cont. | Exp. | Temp. |
|---|---|---|---|---|
| BMGF-RoBERTa (Liu et al., 2020) | 59.44 | 60.98 | 77.66 | 50.26 |
| CG-T5 (Jiang et al., 2021) | 55.40 | 57.04 | 74.76 | 41.54 |
| GOLF (Jiang et al., 2022) | 67.71 | 62.90 | 79.41 | 54.55 |
| DiscoPrompt (Chan et al., 2023b) | 62.55 | 64.45 | 78.77 | 57.41 |
| ContrastiveIDRR (Long and Webber, 2022) | 65.84 | 63.55 | 79.17 | **69.86** |
| PLSE | **70.93** | **67.04** | **81.50** | 66.13 |

Table 3: The performance for top-level classes on PDTB 2.0 in terms of F1 score (%).

and 2.31% second-level accuracy over the current state-of-the-art ContrastiveIDRR model (Long and Webber, 2022) on PDTB 2.0. Moreover, with regard to CoNLL16, our method also outperforms the current state-of-the-art DiscoPrompt model (Chan et al., 2023b) by a margin of 8.70% F1 on CoNLL-Test and 0.47% accuracy on CoNLL-Blind. For the performance of top-level classes on PDTB 2.0, our method significantly improves the performance of senses, particularly in *Comp*, *Cont* and *Exp*. It surpasses the previous best results by 3.22%, 2.59% and 2.09% F1, respectively, demonstrating a substantial efficiency of our method. For the performance of second-level types on PDTB 2.0, we also compare our model with the current state-of-the-art models. As illustrated in Table 4, our results show a noteworthy enhancement in F1 performance across five categories of second-level senses. Meanwhile, our model still remain exceptional average performance in most second-level senses.

**Secondly**, a series of recent works for IDRR (Wu et al., 2022; Jiang et al., 2022; Chan et al., 2023b; Long and Webber, 2022) frequently utilize the hierarchical structure information from the annotated senses, leading to good results. Nevertheless, it's crucial to note that the performance and robust-

| Second-level Sense | GOLF | DP | Contrast | ours |
|---|---|---|---|---|
| Comp.Concession | 0.00 | **9.09** | 0.00 | 0.0 |
| Comp.Contrast | 61.95 | 59.26 | **62.63** | 60.28 |
| Cont.Cause | 65.35 | 63.83 | 65.58 | **68.36** |
| Cont.Pragmatic Cause | 0.00 | 0.00 | 0.00 | 0.00 |
| Exp.Alternative | 63.49 | **72.73** | 53.85 | 60.87 |
| Exp.Conjunction | 60.28 | **61.08** | 58.35 | 59.80 |
| Exp.Instantiation | 75.36 | 69.96 | 73.04 | **76.71** |
| Exp.List | 27.78 | 37.50 | 34.78 | **40.00** |
| Exp.Restatement | 59.84 | 60.00 | 58.45 | **63.78** |
| Temp.Asynchronous | 63.82 | 57.69 | 59.79 | **64.22** |
| Temp.Synchrony | 0.00 | 0.0 | **78.26** | 57.14 |

Table 4: The performance for second-level types on PDTB 2.0 in terms of F1 score (%). **DP** and **Contrast** indicate the DiscoPrompt and ContrastiveIDRR.

ness of these models are excessively reliant on the small-scale annotated data, that may pose a stumbling block to future research explorations. In the NLP community, it has become a prevalent trend to leverage large-scale unlabeled data and design superior self-supervised learning methods. Therefore, our work focuses on sufficiently exploiting unannotated data containing explicit connectives through prompt-based connective prediction. At the same time, to tackle the local dependencies of predicted connectives representations, we design a MI-maximization strategy, which aids in capturing the global logical semantics information pertinent to the predicted connectives. Overall results demonstrate the great potential of our method.

**Finally**, we compare our method with ChatGPT (Chan et al., 2023a), a recent large language model, on the IDRR task. The performance of ChatGPT significantly trails our PLSE model by approxi-

| Model | Top-level | | Second-level | |
|---|---|---|---|---|
| | $F1$ | $Acc.$ | $F1$ | $Acc.$ |
| PLSE | **71.40** | **75.43** | **50.11** | **64.00** |
| w/o $\mathcal{L}_{GLSL}$ -$MTL_{cls}$ | 68.64 | 73.23 | 47.53 | 63.52 |
| w/o $\mathcal{L}_{GLSL}$ -$MTL_{mean}$ | 69.68 | 74.87 | 48.76 | 63.43 |
| w/o $\mathcal{L}_{GLSL}$ | 69.17 | 74.66 | 47.94 | 63.71 |
| w/o $\mathcal{L}_{CM}$ | 69.61 | 73.13 | 48.71 | 63.61 |
| w/o $\mathcal{L}_{MLM}$ | 70.12 | 74.57 | 48.63 | 63.43 |
| w/o $\mathcal{L}_{GLSL}, \mathcal{L}_{CM}$ | 68.18 | 73.23 | 47.20 | 63.13 |
| w/o $\mathcal{L}_{GLSL}, \mathcal{L}_{MLM}$ | 68.62 | 74.95 | 47.50 | 63.62 |
| w/o $\mathcal{L}_{GLSL}, \mathcal{L}_{CM}, \mathcal{L}_{MLM}$ | 67.12 | 71.03 | 46.54 | 62.17 |

Table 5: Ablation study on PDTB 2.0 in terms of Macro-F1 score (%) and Accuracy (%); $MTL_{cls}$ and $MTL_{mean}$ stand for the multi-task learning method that additionally utilizes the [CLS] representation or the mean representation of a sequence to predict senses.

mately 35% F1 and 30% accuracy on PDTB 2.0, which highlights ChatGPT's poor capacity to comprehend the logical semantics in discourses. Therefore, further research on IDRR remains crucial and imperative for the NLP community.

## 5.2 Ablation Study

To better investigate the efficacy of individual modules in our framework, we design numerous ablations on *Global Logical Semantics Learning*, *Connectives Mask* and *MaskedLM* tasks. Table 5 indicates that eliminating any of the three tasks would hurt the performance of our model on both top-level and second-level senses. It is worth noting that removing *Global Logical Semantics Learning* significantly hurts the performance. Furthermore, to assess the proficiency of *Global Logical Semantics Learning*, we compare it with an auxiliary classification task leveraging the [CLS] representation or the average representation of a sequence, which aims to introduce global information. The performance's improvement through $MTL_{mean}$ highlights the effectiveness of incorporating global semantics. Surprisingly, $MTL_{cls}$ detrimentally impacts the model's performance. The reason could be that the [CLS] representation encompasses a vast amount of information irrelevant to global logical semantics, thereby interfering with connective prediction. Our method considerably outperforms $MTL_{mean}$, which demonstrates that our MI maximization strategy by integrating global logical semantics is indeed beneficial for connective prediction. Besides, eliminating *Connectives Mask* or *MaskedLM* also diminishes the performance of our model, illustrating that our method can effectively utilize large-scale

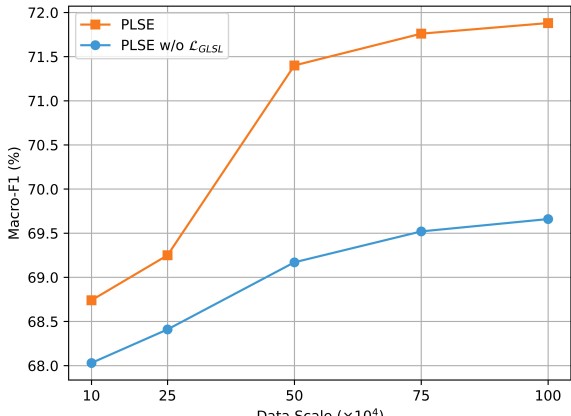

Figure 3: Effects of the unannotated data scale on the test set in PDTB 2.0.

unannotated data to learn better discourse relation representations by injecting discourse logical semantics and universal semantics into PLMs.

## 5.3 Data Scale Analysis

To investigate the impact of unannotated data scale on the performance of our model, we conduct supplementary extension experiments. As show in Figure 3, the performance swiftly accelerates prior to reaching a data scale of $500,000$, yet beyond this threshold, the pace of improvement markedly decelerates. The primary reason is the difference in semantic distributions of explicit and implicit data. For example, the connectives in explicit data play an essential role in determining the discourse relations. Conversely, the implicit data does not contain any connectives, while the discourse relations of sentences remain unaffected by implicit connectives in annotations. Therefore, considering both performance and training cost, we select unannotated explicit data at a scale of $500,000$.

## 5.4 Few-Shot Learning

To evaluate the robustness of our model in few-shot learning, we conduct additional experiments using 10%, 20%, and 50% of training data, respectively. The results of few-shot learning are summarized in Figure 4. The orange line illustrates the outcomes achieved by our PLSE method, while the blue line depicts the results obtained without pretraining. The results of the comparison demonstrate that our method significantly outperforms the baseline across all metrics. In particular, our method, when trained on only 50% of training data, achieves performance comparable to the baseline trained on full training data, which further underscores the efficacy of our approach.

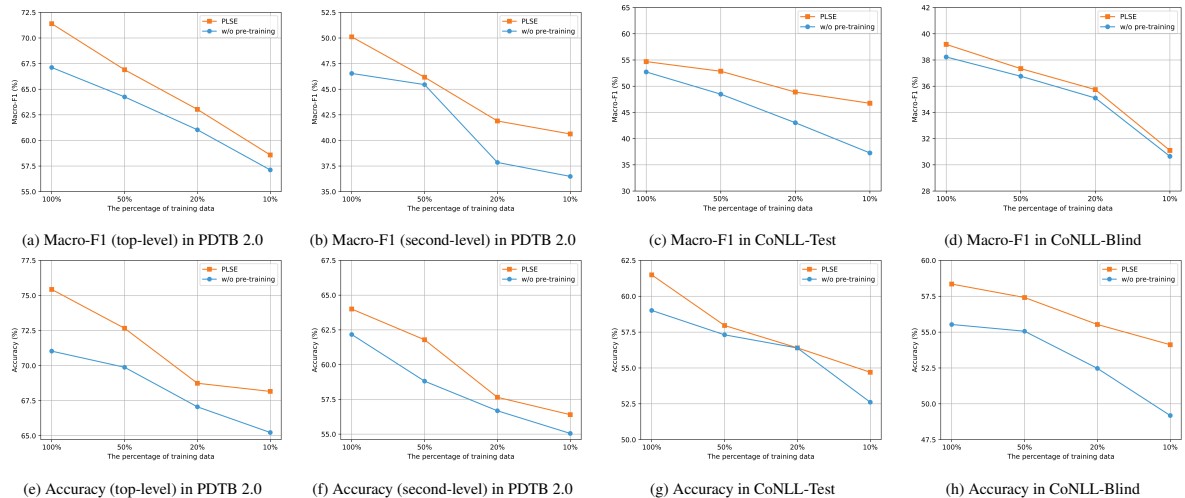

Figure 4: Performance comparison of few-shot learning on PDTB 2.0 and CoNLL16 datasets.

# 6 Conclusion

In this paper, we propose a novel prompt-based logical semantics enhancement method for IDRR. Our approach sufficiently exploits unannotated explicit data by seamlessly incorporating discourse relation knowledge based on *Cloze-Prompt* connective prediction and comprehensively learning global logical semantics through MI maximization. Compared with the current state-of-the-art methods, our model transcends the limitations of overusing annotated multi-level hierarchical information, which achieves new state-of-the-art performance on PDTB 2.0 and CoNLL2016 datasets.

# Limitations

Despite achieving outstanding performance, there are still some limitations of our work, which could be summarized into the following two aspects. The first is the different semantic distributions between explicit and implicit data, as illustrated in Section 5.3, that significantly hampers further improvements of our model's performance. A promising future research would involve the exploration of innovative approaches to bridging the gap between explicit and implicit data, or the development of strategies to filter explicit data exhibiting similar distribution patterns to implicit data. Additionally, our connective prediction method is limited by its reliance on single tokens, thereby transforming two-token connectives into one (e.g., *for example* → *example*). Moreover, it discards three-token connectives entirely (e.g., *as soon as*). As a result, our approach doesn't fully exploit the pre-trained models and explicit data to some extent. In the future, we will explore new methods for predicting multi-token connectives.

# Ethics Statement

This study focuses on model evaluation and technical improvements in fundamental research. Therefore, we have refrained from implementing any additional aggressive filtering techniques on the text data we utilize, beyond those already applied to the original datasets obtained from their respective sources. The text data employed in our research might possess elements of offensiveness, toxicity, fairness, or bias, which we haven't specifically addressed since they fall outside the primary scope of this work. Furthermore, we do not see any other potential risks.

# Acknowledgement

The authors would like to thank the organizers of EMNLP2023 and the reviewers for their helpful suggestions. This work is partly supported by the grants from the National Natural Science Foundation of China (No. 62172044) and the Open Project Program of the National Defense Key Laboratory of Electronic Information Equipment System Research (No. 614201001032203).

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

# A  Appendix

## A.1  Multi-level Sense Hierarchy

The hierarchies of both PDTB 2.0 and CoNLL16 consist of three levels. According to the PDTB annotation guidelines, all senses are organized into a three-layer hierarchical structure. Figure 5 shows the multi-level sense hierarchy of an IDRR instance in the PDTB 2.0 corpus.

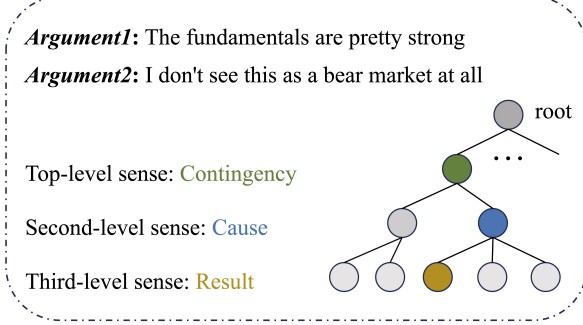

Figure 5: The multi-level sense hierarchy of an IDRR instance in the PDTB 2.0 corpus.

## A.2  Data Statistics

| Top-level Senses | Train | Dev | Test |
|---|---|---|---|
| Comparison (Comp.) | 1942 | 197 | 152 |
| Contingency (Cont.) | 3340 | 292 | 279 |
| Expansion (Exp.) | 7004 | 671 | 574 |
| Temporal (Temp.) | 760 | 64 | 85 |
| Total | 12632 | 1183 | 1046 |

Table 6: Data statistics of four top-level implicit senses in PDTB 2.0.

| Second-level Senses | Train | Dev | Test |
|---|---|---|---|
| Comp.Concession | 184 | 15 | 17 |
| Comp.Contrast | 1610 | 171 | 134 |
| Cont.Cause | 3277 | 284 | 272 |
| Cont.Pragmatic cause | 64 | 7 | 7 |
| Exp.Alternative | 151 | 10 | 9 |
| Exp.Conjunction | 2882 | 264 | 209 |
| Exp.Instantiation | 1102 | 108 | 122 |
| Exp.List | 338 | 10 | 12 |
| Exp.Restatement | 2458 | 271 | 216 |
| Temp.Asynchronous | 555 | 50 | 57 |
| Temp.Synchrony | 204 | 13 | 28 |
| Total | 12406 | 1165 | 1039 |

Table 7: Data statistics of 11 second-level implicit senses in PDTB 2.0.

## A.3  Verbalizer for CoNLL16

| Cross-level Senses | Connectives |
|---|---|
| Comp.Concession | although, however |
| Comp.Contrast | but |
| Cont.Cause.Reason | because |
| Cont.Cause.Result | so, consequently, thus, therefore |
| Cont.Condition | if |
| Exp.Alternative | unless |
| Exp.Alternative.Chosen alternative | instead, rather |
| Exp.Conjunction | and, also, fact, furthermore |
| Exp.Exception | except |
| Exp.Instantiation | instance, example |
| Exp.Restatement | particular, indeed, specifically |
| Temp.Asynchronous.Precedence | then, before |
| Temp.Asynchronous.Succession | after |
| Temp.Synchrony | meanwhile, when |

Table 8: The verbalizer mapping between implicit discourse relation labels and connectives on CoNLL16 dataset, including 14 cross-level senses.

### A.4  Mutual Information Discriminator

We concatenate the global logic-related representation $H_{Glogic}$ and the connective-related representation $H_{cmask}$ as input. Then we feed this to the discriminator $T_\omega$, which produces scores to maximize the MI estimator in Equation (5). The architecture of $T_\omega$ is shown in Table 9.

| Operation | Size | Activation |
|---|---|---|
| Input $\rightarrow$ Linear layer | $1536 \rightarrow 768$ | ReLU |
| Linear layer | $768 \rightarrow 768$ | ReLU |
| Linear layer | $768 \rightarrow 1$ | |

Table 9: The network architecture of discriminator $T_\omega$ .