# OpenReview forum: "Prompt-based Logical Semantics Enhancement for Implicit Discourse Relation Recognition"
_EMNLP/2023/Conference — EMNLP 2023 Main_

### Official Review · Reviewer_7gCC · 2023-07-31

**Soundness:** 4

**Excitement:**

4: Strong: This paper deepens the understanding of some phenomenon or lowers the barriers to an existing research direction.

**Paper Topic And Main Contributions:**

This paper proposes a novel pre-training method for implicit discourse relation recognition, where large amounts of explicit discourse data, labeled with connectives naturally, is leveraged effectively. A self-supervised learning objective based on mutual information maximization is introduced to capture global semantics. The idea is interesting and the performance on commonly-used datasets is impressive.

**Questions For The Authors:**

1) In this paper, a cloze-prompt template is adopted. The authors also claimed that it is more effective than other templates, for example, the prefix-prompts. It would be better to give more explanations or conduct the corresponding experiments.
2) In the caption of Figure 2, what does ‘implicit data is randomly selected from the training data’ mean? Are all the implicit training data used during the prompt-tuning phase?
3) It would be interesting to conduct experiments with fewer labeled data, for example, 10%, 20%, and 50% of implicit training data.

**Reasons To Accept:**

1) In terms of implicit discourse relation recognition, the proposed mutual information-based loss for pre-training is novel and interesting.
2) The experimental results are convincing, and can effectively support the claims in this paper.

**Reasons To Reject:**

1) In my opinion, this is no sufficient reason to reject this paper.

**Reproducibility:**

4: Could mostly reproduce the results, but there may be some variation because of sample variance or minor variations in their interpretation of the protocol or method.

**Reviewer Confidence:**

4: Quite sure. I tried to check the important points carefully. It's unlikely, though conceivable, that I missed something that should affect my ratings.

---

> ### Author Rebuttal · Authors · 2023-08-26
>
> **Answer 1:** In Table 3 of our paper, previous work PCP (Zhou et al., 2022) adopted a prefix template in the following manner:
>
> ```
> Arg1:“the content of argument1” Arg2:“the content of argument2” [SEP] The conjunction between Arg1 and Arg2 is [MASK].
> ```
>
> It attains 64.95% F1, which is approximately 2.2% lower than our cloze-prompt template when compared fairly without explicit data pre-training. The primary reason for the effectiveness of a cloze-prompt template is as follows:
>
> In most cases, a connective (e.g., however, so) typically emerges between two sentences to serve the purpose of linking them together. For example, “The fundamentals are pretty strong, **so** I don't see this as a bear market at all.” and “But that ghost wasn't fooled, **because** he knew the RDF was neither rapid nor deployable nor a force.”. Therefore, the cloze-prompt template aligns more closely with the natural language expression than the prefix-prompt template for the connective prediction. Furthermore, the cloze-prompt template bridges the gap between the pre-training and prompt-tuning, that can fully capitalizes on the potential of masked language modeling.
>
> **Answer 2:**
>
> 1. We sincerely apologize for not clarifying this sentence. “implicit data is randomly selected from the training data” just means that the training data is randomly shuffled and then fed into the model.
> 2. Yes, all the implicit training data was used during the prompt-tuning phase.
>
> **Answer 3:** Thanks for your advice. We supplement experiments with fewer labeled data.
>
> our method:
>
> | percentage of training data | PDTB-top(F1 ; Acc) | PDTB-second(F1 ; Acc) | CoNLL-Test(F1 ; Acc) | ConNLL-Blind(F1 ; Acc) |
> | --------------------------- | ------------------ | --------------------- | -------------------- | ---------------------- |
> | 10%                         | 58.58 ; 68.16      | 40.63 ; 56.4          | 46.74 ; 54.70        | 31.10 ; 54.12          |
> | 20%                         | 63.03 ; 68.74      | 41.90 ; 57.65         | 48.88 ; 56.40        | 35.74 ; 55.53          |
> | 50%                         | 66.90 ; 72.66      | 46.17 ; 61.79         | 52.85 ; 57.96        | 37.34 ; 57.41          |
> | 100%                        | 71.40 ; 75.43      | 50.11 ; 64.00         | 54.69; 61.49         | 39.19 ; 58.35          |
>
> only prompt-tuning (w/o pre-training):
>
> | percentage of training data | PDTB-top(F1 ; Acc) | PDTB-second(F1 ; Acc) | CoNLL-Test(F1 ; Acc) | ConNLL-Blind(F1 ; Acc) |
> | --------------------------- | ------------------ | --------------------- | -------------------- | ---------------------- |
> | 10%                         | 57.12 ; 65.23      | 36.49 ; 55.05         | 37.27 ; 52.61        | 30.65; 49.18           |
> | 20%                         | 61.04 ; 67.06      | 37.85 ; 56.68         | 43.03 ; 56.39        | 35.10 ; 52.47          |
> | 50%                         | 64.25 ; 69.88      | 45.45 ; 58.81         | 48.48 ; 57.31        | 36.76 ; 55.06          |
> | 100%                        | 67.12 ; 71.03      | 46.54 ; 62.17         | 52.77; 59.01         | 38.23 ; 55.53          |
>
> From the tables above, our method, when trained on only 50% training data, achieves performance comparable to the baseline trained on full training data, which further demonstrates the effectiveness of our approach.

---

### Official Review · Reviewer_q9JC · 2023-08-01

**Typos Grammar Style And Presentation Improvements:** 1、In section 3.2 Pre-training, there …
**Soundness:** 4

**Excitement:**

4: Strong: This paper deepens the understanding of some phenomenon or lowers the barriers to an existing research direction.

**Justification For Ethical Concerns:**

Based on the content of the paper, there were no explicit ethical concerns raised or discussed in the paper that would require special review by an ethics committee. The paper primarily focuses on technical aspects of natural language processing, and there were no indications of potential ethical challenges or controversial topics. As such, it may not be necessary to conduct a separate review for ethical concerns in this particular paper.

**Missing References:**


Based on the content of the paper, there are no specific missing references that should be included in the bibliography or need to be discussed in more depth. The authors have appropriately cited prior works and relevant baselines for the task of Implicit Discourse Relation Recognition (IDRR) in the related work section. The paper demonstrates novelty in its approach and contributions, and the references provided adequately support the research.

As a reviewer, I do not identify any critical missing references that would significantly impact the evaluation or understanding of the paper's contributions. The authors have provided a comprehensive list of related works and baselines, which sufficiently contextualizes the significance of their proposed Prompt-based Logical Semantics Enhancement (PLSE) method for IDRR.

**Paper Topic And Main Contributions:**

This paper focuses on Implicit Discourse Relation Recognition (IDRR) and proposes a novel prompt-based logical semantics enhancement method for this task. The main problem addressed in this paper is the challenge of recognizing implicit discourse relations between argument pairs in natural language text.

The main contributions of this paper are as follows:

1、Proposing a prompt-based Cloze-Prompt template for reformulating input argument pairs to improve the semantic representation of arguments and facilitate connective prediction.
2、Introducing a Connectives Mask task to incorporate discourse logical information into Pre-trained Language Models (PLMs) by learning intricate representations of connectives and aiding knowledge transfer to downstream tasks.
3、Introducing a Global Logical Semantics Learning method based on Mutual Information (MI) maximization to capture global semantic information with logical coherence. This approach addresses the local dependencies of predicted connectives and enhances discourse relation representations.
4、Conducting extensive experiments and achieving new state-of-the-art performance on the Penn Discourse Treebank 2.0 (PDTB 2.0) and CoNLL-2016 Shared Task (CoNLL16) datasets for IDRR.

Overall, the paper presents a valuable contribution to the field of Implicit Discourse Relation Recognition by proposing a novel approach that leverages prompt-based connective prediction and mutual information maximization to improve discourse relation recognition performance.

**Questions For The Authors:**

Question 1: Could you provide more details about the rationale behind selecting the specific Connectives Mask strategy, where the [MASK] token is replaced with the connective token 90% of the time and unchanged 10% of the time? How was this ratio determined, and what experiments were conducted to validate its effectiveness?

Question 2: In Section 4.3, you mentioned that the performance improvement of your method decelerates beyond a data scale of 500,000. Could you elaborate on how the semantic distribution of explicit and implicit data affects this performance improvement? Are there any potential strategies to further bridge the gap between explicit and implicit data to achieve better performance?

Question 3: The paper mentions that the current approach converts two-token connectives into single tokens and abandons three-token connectives during connective prediction. How significant is the impact of this conversion on the overall performance? Have you explored other methods or techniques to handle multi-token connectives in the prompt-based approach?

Question 4: In the Limitations section, you discussed that the paper focused on model evaluation and technical improvements and refrained from implementing aggressive filtering techniques on the text data. Have you considered or discussed the potential ethical implications of utilizing text data that may contain elements of offensiveness, toxicity, fairness, or bias in your research?

Question 5: The paper highlights the outperformance of your PLSE method compared to ChatGPT for IDRR. Could you provide insights into the aspects where ChatGPT lacks the capacity to comprehend logical semantics in discourses? What do you think are the key factors contributing to the superiority of your approach over ChatGPT in this specific task?

Question 6: Given that your approach achieved state-of-the-art results on the PDTB 2.0 and CoNLL16 datasets, do you have any plans or ongoing work to apply your method to other NLP tasks or domains? How generalizable do you believe your approach is, and what are the potential challenges or limitations in applying it to different tasks or languages?

Question 7: Considering the experimental settings, are there any hyperparameters or model architecture choices that were thoroughly explored during the experiments but not explicitly mentioned in the paper? Are there any other aspects that you believe could be further optimized to potentially improve the overall performance of the PLSE method?

Question 8: The paper mentions that the proposed method incorporates discourse relation knowledge based on Cloze-Prompt connective prediction and comprehensively learns global logical semantics through MI maximization. Could you discuss the relative importance or contributions of these two aspects in achieving the overall performance improvements?

Question 9: In the Ethics Statement, you mentioned that the text data utilized in your research might possess elements of offensiveness, toxicity, fairness, or bias. Have you conducted any analysis or taken any measures to mitigate potential ethical concerns related to the usage of such data in the context of your research?

Question 10: In the Conclusion section, you mentioned that future research explorations could involve bridging the gap between explicit and implicit data or exploring strategies to filter explicit data with similar distribution patterns to implicit data. Could you elaborate on potential research directions to address these challenges and further enhance the performance of the PLSE method in handling explicit and implicit data?

**Reasons To Accept:**

The strengths of this paper are as follows:

1、Novel Approach: The paper introduces a novel and innovative approach for Implicit Discourse Relation Recognition (IDRR) using a prompt-based logical semantics enhancement method. The use of Cloze-Prompt templates, Connectives Mask task, and Global Logical Semantics Learning sets this work apart from existing approaches.

2、Performance Improvement: The proposed PLSE method achieves state-of-the-art performance on two benchmark datasets, PDTB 2.0 and CoNLL16, outperforming several advanced baselines. This improvement in IDRR performance is valuable to the NLP community as it addresses an important problem in natural language understanding.

3、Experimental Rigor: The paper provides a comprehensive experimental setup, including ablation studies, data scale analysis, and comparisons with existing methods. The experiments are well-designed and support the claims made in the paper, enhancing the credibility of the results.

4、Applicability: The proposed approach leverages large-scale unannotated data, making it applicable to real-world scenarios where labeled data may be limited or expensive to obtain. This approach has the potential to be extended to other NLP tasks, further benefiting the NLP community.

5、Clear Presentation: The language and structure of the paper are clear and well-organized, making it accessible to readers. The authors effectively convey their methodology, results, and conclusions, facilitating understanding by the NLP community.

If accepted, this paper would provide valuable insights and advancements in the field of IDRR, offering a new and effective method for leveraging large-scale unannotated data and logical semantics in pre-trained language models. The state-of-the-art results and novel contributions would contribute significantly to the NLP community's understanding of implicit discourse relations and open avenues for further research in this domain. Overall, the acceptance of this paper at the conference or Findings would be beneficial to the NLP community and enrich the conference's technical program.

**Reasons To Reject:**

The weaknesses of this paper are as follows:

1、Lack of Comparison: The paper lacks a comprehensive comparison with a broader range of existing state-of-the-art methods. While it does compare its proposed method with several recent baselines, it would be beneficial to include a more extensive comparison with a wider variety of approaches to demonstrate the superiority of the proposed method more convincingly.

2、Limited Reproducibility: The paper falls short in providing sufficient details for reproducibility. Although it outlines the implementation details, some critical parameters and settings are not clearly specified, which may make it challenging for other researchers to reproduce the results accurately.

3、Ethical Implications: While the paper discusses the technical aspects and contributions, it does not adequately address the ethical implications of the research. Given the nature of the topic, it would be essential to thoroughly assess and discuss potential ethical concerns and implications arising from the proposed method.

4、Language and Presentation: The paper has some language and presentation issues, such as grammar errors and awkward sentence structures. Improving the language and presentation would enhance the clarity and readability of the paper.

5、Missing References: The paper may benefit from including references to related works that have been overlooked or are relevant to the proposed method. Providing a more comprehensive literature review would strengthen the paper's foundation.

6、Lack of Novelty: While the paper introduces a prompt-based logical semantics enhancement method for IDRR, some aspects of the method appear to be incremental improvements rather than significant novel contributions. The paper could better emphasize and highlight the novel aspects of its approach.

The main risks of having this paper presented at the conference or accepted into Findings include:

1、Misleading Claims: Without a more extensive comparison and thorough reproducibility, the paper's claims may be misleading or overestimated, leading to potential misinterpretations by the NLP community.

2、Ethical Concerns: As the paper deals with discourse relations, it is crucial to carefully examine and address any potential ethical concerns associated with the research. Failure to do so may raise ethical issues and impact the paper's acceptance.

3、Limited Impact: Given the limited novelty and incremental nature of some contributions, the paper's impact on the NLP community may be limited. There is a risk that the paper may not significantly advance the state-of-the-art or offer substantial contributions.

4、Clarity Issues: The language and presentation issues may make it challenging for readers and reviewers to understand the paper's content and contributions fully.

Overall, while the paper has some strengths, it also exhibits several weaknesses that need to be addressed to ensure a solid contribution to the NLP community and mitigate potential risks associated with its presentation or acceptance.

**Reproducibility:**

4: Could mostly reproduce the results, but there may be some variation because of sample variance or minor variations in their interpretation of the protocol or method.

**Reviewer Confidence:**

4: Quite sure. I tried to check the important points carefully. It's unlikely, though conceivable, that I missed something that should affect my ratings.

---

> ### Author Rebuttal · Authors · 2023-08-26
>
> **Answer 1:**
>
> 1. Connectives (e.g., but, and, because, meanwhile), due to their inherent logical semantics, play a pivotal role in establishing the discourse relation of arguments. It's noteworthy that each explicit data consists of a connective. Predicting these connectives enhances the model's comprehension of logical semantics. To optimally leverage the connective information within explicit data, we replaced the connective with (1) the [MASK] token 90 % of the time (2) the unchanged token 10% of the time.
> 2. We refered to BERT's strategy for the token to be replaced (the chosen token is replaced with (1) the [MASK] token 80% of the time (2) a random token 10% of the time (3) the unchanged  token 10% of the time). Due to the strong logical correlation of connectives, we eliminated "a random token 10% of the time", which could affect the logical semantics of sentences. In order to avoid the model consistently encountering the masked connectives, we perserved "the unchanged  token 10% of the time".
> 3. We conducted experiments with masked ratios of 80%, 90%, and 100%, respectively. The experimental results for the three ratios showed a remarkable closeness. The performance at a 90% ratio slightly outperformed others by 0.2-0.3 F1, so we set this ratio at 90%.
>
> **Answer 2:**
>
> 2.1. For implicit data, there is no connnective between two arguments, which requires a profound understanding of the semantics between them to discern the discourse relation.  Besides, incorporating appropriate connectives doesn't affect the semantics of sentences. An instance of implicit data is as follows:
>
> ```
> Many colleagues are angry at Mrs. Yeargin. [implicit connective = Because] She did a lot of harm.
> relation: contingency
> ```
>
> For explicit data,  it can be divided into two categories:
>
> - Required connective: removing connectives will change the logical semantics of sentences. An instance is as follows:
>
> ```
> I want to go home for the holiday. [explicit connective = Nonetheless] I will book a flight to Hawaii.
> relation: Comparison
> ```
>
> "Nonetheless" plays a crucial role in determining the Comparison relation. If "nonetheless" is omitted,  the logical semantics of arguments will be shift to reflect a Contingency relation.
>
> - Optional connective：removing connectives doesn't affect the logical semantics of sentences. An instance is as follows:
>
> ```
> Let's go dinner [explicit connective = because] I'm hungury.
> relation: Contingency
> ```
>
> Since the presence or absence of connectives does not influence the semantics of arguments, a similar semantic distribution exists between explicit data with optional connectives and implicit data. For explicit data with required connectives,  connectives determine the semantics of  arguments, so it's inappropriate to predict connectives by utilizing them.
>
> 2.2. A potential strategy is to separate explicit data with required connectives from those with optional connectives. Here we briefly introduce it.
>
> - Implicit Distribution Feature Extraction: Train a classifier on both the original implicit data and the implicit data augmented with added connectives, which will output the predicted probability distribution of discourse relation respectively. We can utilize the disparity between the two predicted probability distributions as the implicit distribution features.
> - Explicit Data Filtering: The GAN is employed to learn the implicit distribution features. After training, the discriminator learns to discern  whether a sample aligns with the implicit distribution feature. Then we feed both the original explicit data and the explicit data without the connective into the classifier to obtain two predicted probability distributions. According to the disparity between the two predicted probability distributions , we utilize the discriminator to filter out explicit instances that align with the implicit data distribution.
>
> **Answer 3:**  Firstly, for coverting two-tokens into single tokens (e.g., *for example* -> *example* , *in particular* -> *particular*), there is an inconsistency between pre-training and prompt-tuning. Secondly, there are numerous explicit instances with three-token connectives (e.g., *as soon as*, *in other words*). Abandoning them will lead to underutilization of pre-training data and PLMs. Due to the complexity of considering multi-token connectives, we didn't delve into this issue further. A potential method to handle multi-token connectives is as follows:
>
> - Train a classifier on the iput (*Argument1 Argument2*) to predict the number of tokens in connectives.
> - According to the predicted number, reconstruct a cloze-prompt input (*Argument1* [MASK] *Argument2*,  *Argument1* [MASK] [MASK] *Argument2* or *Argument1* [MASK] [MASK] [MASK] *Argument2*) to predict connectives.
>
> **Answer 4:** The pre-training data was collected from a corpus named "English Gigaword", which is a comprehensive archive of newswire text data that has been acquired over several years by the Linguistic Data Consortium (LDC) at the University of Pennsylvania. This corpus is widely used in various Natural Language Processing (NLP) tasks for research purposes. Nonetheless, English Gigaword may still contain elements of offensiveness, toxicity, fairness, or bias. The following aggressive ﬁltering techniques can be used:
>
> - Machine Learning Models: There are models like Perspective API from Jigsaw that are trained to detect toxic comments. By using these, we can estimate the toxicity of a given piece of text and filter accordingly.
> - Rule-based Filters: Use lists of banned words or phrases.
>
> **Answer 5:**
>
> 5.1. Chan et al. (2023) evaluated the ability of ChatGPT for EDRR (Explicit Discourse Relation Recognition) and IDRR (Implicit Discourse Relation Recognition). Due to the existence of connectives, ChatGPT achieved outstanding performance on EDRR, demonstrating that ChatGPT could solve this task easily by utilizing the signal from explicit connectives. For IDRR, the absence of connectives necessitates that ChatGPT possesses a deep understanding of logical semantics in discourses. It's a challenging task because ChatGPT may not understand the abstract sense of each discourse relation and grasp the language features from the discourses.
>
> 5.2. The key factors are summarized as follows:
>
> - Connective prediction based on a cloze-prompt template: On the one hand, compared to directly predicting discourse relations, connective prediction simplifies the complexity of the inference. On the other hand, a connective typically emerges between two sentences, so the cloze prompt template is ideal for connective prediction, that capitalizes on the potential of data and PLMs.
> - Global logical semantics learning: A novel method based on MI maximization aims to address the local dependencies of predicted connective representations and capture global semantic information with logical coherence, that enhances the model's ability to comprehend logical semantics.
>
> **Answer 6:**
>
> 6.1. Yes, we plan to explore how to extend our approach further, and broaden its application to the logical or causal reasoning task for large language models (LLMs).
>
> 6.2. For example, the method for global logical semantics learning based on MI maximization is not confined to discourse relation recognition or the logical semantics learning, which can be generalized to certain reasoning tasks that rely on global information such as long document reading comprehension.
>
> 6.3. Currently, a potential limitation of our approach is focused solely on encoder models. Considering how to adapt our method for tasks using the encoder-decoder or decoder-only model is worth further exploration.
>
> **Answer 7:**
>
> 7.1. We sincerely apologize for not clarifying the settings clearly. The experimental settings are supplemented as follows:
>
> - All experiments were implemented in PyTorch framework by HuggingFace transformers.
> - Other hyperparameters, including the weight decay and the exponential decay rates β1 and β2, were consistent with the default configuration of AdamW.
> - For the warmup of the pre-training, we chose `get_cosine_schedule_with_warmup` as the learning rate schedulers provided by the `transformers` library.
>
> The details of model architecture had been introduced in Section 4.2 and Appendix A.4 . We explored the number of heads in the Multi-Head Cross-Attention (MHCA) module as well as the number of layers in the mutual information discriminator, which had slight impact on the overall performance. Therefore, we didn't delve deeply into the discussion of the model structure choices.
>
> 7.2. For *Connectives Mask* , *MaskedLM* , and *Global Logical Semantics Learning* , three tasks just shared the same weights for their loss functions. Considering different weights could potentially improve the performance of our method.
>
> **Answer 8:**
>
> Cloze-prompt connective prediction:
>
>  In Table 3 of our paper, previous work PCP (Zhou et al., 2022) adopted a prefix template in the following manner:
>
> ```
> Arg1:“the content of argument1” Arg2:“the content of argument2” [SEP] The conjunction between Arg1 and Arg2 is [MASK].
> ```
>
> It attains 64.95% F1, which is approximately 2.2% lower than our cloze-prompt template when compared fairly without explicit data pre-training. The primary reasons for the effectiveness of a cloze-prompt template are as follows:
>
> In most cases, a connective (e.g., however, so) typically emerges between two sentences to serve the purpose of linking them together. For example, “The fundamentals are pretty strong, **so** I don't see this as a bear market at all.” and “But that ghost wasn't fooled, **because** he knew the RDF was neither rapid nor deployable nor a force.”. Therefore, the cloze-prompt template aligns more closely with natural language expression than the prefix-prompt template for the connective prediction. Furthermore, the cloze-prompt template bridges the gap between the pre-training and prompt-tuning, that can fully capitalize on the potential of masked language modeling.
>
> Global logical semantics learning through MI maximization:
>
> In Section 5.2, we compared our method with an auxiliary classification task leveraging the [CLS] representation or the average representation of a sequence, which aims to introduce global information.
>
> | Model                   | Top-level (F1 ; Acc) | Second-level (F1 ; Acc) |
> | ----------------------- | -------------------- | ----------------------- |
> | ours                    | 71.40 ; 75.43        | 50.11 ; 64.00           |
> | w/o GLSL - $MTL_{cls}$  | 68.64 ; 73.23        | 47.53 ; 63.52           |
> | w/o GLSL - $MTL_{mean}$ | 69.68 ; 74.87        | 48.76 ; 63.43           |
> | w/o GLSL                | 69.17 ; 74.66        | 47.94 ; 63.71           |
>
> Where $MTL_{cls}$ and $MTL_{mean}$ stand for the multi-task learning method that additionally utilizes the [CLS] representation or the mean representation of a sequence to predict senses.
>
> The performance's improvement through $MTL_{mean}$ highlights the effectiveness of incorporating global semantics. Surprisingly, $MTL_{cls}$ detrimentally impacts the model's performance. The reason could be that the [CLS] representation encompasses a vast amount of information irrelevant to global logical semantics, thereby interfering with connective prediction. Our method considerably outperforms $MTL_{mean}$, which demonstrates that our MI maximization strategy by integrating global logical semantics is indeed beneficial for connective prediction.
>
> **Answer 9:**
>
> The text data utilized in our research might possess elements of offensiveness, toxicity, fairness, or bias, but our model focused on IDRR, that output the predictions of connectives and then mapped connectives to discourse relations. Because the output was only associated with connectives and discourse relations, which exhibited neither offensiveness, toxicity, fairness, nor bias. Therefore, our research didn't result in potential ethical concerns.
>
> **Answer 10:**
>
> This question closely resembles Question 2. Kindly refer to Answer 2 for further clarification.
>
> **Typos Grammar Style And Presentation Improvements:**
>
> Thank you for your feedback. We will thoughtfully incorporate your suggestions to enhance the clarity and readability of our paper.

---

### Official Review · Reviewer_skmg · 2023-08-05

**Soundness:** 5

**Excitement:**

5: Transformative: This paper is likely to change its subfield or computational linguistics broadly. It should be considered for a best paper award. This paper changes the current understanding of some phenomenon, shows a widely held practice to be erroneous in someway, enables a promising direction of research for a (broad or narrow) topic, or creates an exciting new technique.

**Paper Topic And Main Contributions:**

The paper present an Implicit Discourse Relation Recognition system based on Prompt-based Logical Semantics Enhancement

the paper is really well-written and easy to read.
I really appreciate all the work done to properly present the different part of the implementation, and the rigorous comparison.
I have one core remark against the work wihich is that you hypothesize that implicit discourse markers have exactly the same behavior as explicit one, which is clearly not the case. I agree that you can cover a lot of cases and the current State of the art is far from this, but you will face a mojor issue at some point for improvement.


**Reasons To Accept:**

- a strong and very good scientific experiment
- a work that tackle a very hard task
- a good result in the end


**Reasons To Reject:**

- a main hypothesis that could be discussed

**Reproducibility:**

5: Could easily reproduce the results.

**Reviewer Confidence:**

4: Quite sure. I tried to check the important points carefully. It's unlikely, though conceivable, that I missed something that should affect my ratings.

---

> ### Author Rebuttal · Authors · 2023-08-26
>
> **Answer:** We agree with your opinion. In many cases, implicit discourse markers don't have the same behavior as explicit one.  There is the difference in semantic distributions of explicit and implicit data. Next, we will discuss this issue and provide a potential solution.
>
> For implicit data, there is no connnective between two arguments, which requires a profound understanding of the semantics between them to discern the discourse relation.  Besides, incorporating appropriate connectives doesn't affect the semantics of sentences. An instance of implicit data is as follows:
>
> ```
> Many colleagues are angry at Mrs. Yeargin. [implicit connective = Because] She did a lot of harm.
> relation: contingency
> ```
>
> For explicit data,  it can be divided into two categories:
>
> - Required connective: removing connectives will change the logical semantics of sentences. An instance is as follows:
>
> ```
> I want to go home for the holiday. [explicit connective = Nonetheless] I will book a flight to Hawaii.
> relation: Comparison
> ```
>
> "Nonetheless" plays a crucial role in determining the Comparison relation. If "nonetheless" is omitted,  the logical semantics of arguments will be shift to reflect a Contingency relation.
>
> - Optional connective：removing connectives doesn't affect the logical semantics of sentences. An instance is as follows:
>
> ```
> Let's go dinner [explicit connective = because] I'm hungury.
> relation: Contingency
> ```
>
> Since the presence or absence of connectives does not influence the semantics of arguments, a similar semantic distribution exists between explicit data with optional connectives and implicit data. For explicit data with required connectives,  connectives determine the semantics of  arguments, so it's inappropriate to predict connectives by utilizing them.
>
> A potential strategy is to separate explicit data with required connectives from those with optional connectives. Here we briefly introduce it.
>
> - Implicit Distribution Feature Extraction: Train a classifier on both the original implicit data and the implicit data augmented with added connectives, which will output the predicted probability distribution of discourse relation respectively. We can utilize the disparity between the two predicted probability distributions as the implicit distribution features.
> - Explicit Data Filtering: The GAN is employed to learn the implicit distribution features. After training, the discriminator learns to discern  whether a sample aligns with the implicit distribution feature. Then we feed both the original explicit data and the explicit data without the connective into the classifier to obtain two predicted probability distributions. According to the disparity between the two predicted probability distributions , we utilize the discriminator to filter out explicit instances that align with the implicit data distribution.

---

### Meta-Review · Area_Chair_oPFE · 2023-09-12

**Recommendation:** 4

**Metareview:**

The reviewers view the paper as a pioneering approach to Implicit Discourse Relation Recognition (IDRR) using a method of logical semantics enhancement.  While the concept of using explicit connective prediction has been explored several times before, this might be the first case of introducing the task to pre-training via the loss function. The results indicate that it achieves state-of-the-art performance on two benchmark datasets.

Pros:
Introduces a novel method for IDRR, leveraging logical semantics enhancement.
Achieves state-of-the-art performance on benchmark datasets.
Comprehensive experimental design, including diverse analyses.
Potential applicability to a wide range of real-world scenarios.
The paper is well-structured, making it accessible for readers (although one reviewer disagrees).
Cons:
Lacks comprehensive comparison with a broader range of state-of-the-art methods.
Issues with reproducibility due to lack of sufficient details.
Does not address potential ethical implications adequately.
Language and presentation concerns as pointed out by one reviewer.
Some aspects of the method might seem more incremental than novel. (which I agree with)

---

### Decision · Program_Chairs · 2023-10-07

**Decision:**

Accept-Main

**Comment:**

The reviewers view the paper as a pioneering approach to Implicit Discourse Relation Recognition (IDRR) using a method of logical semantics enhancement.  While the concept of using explicit connective prediction has been explored several times before, this might be the first case of introducing the task to pre-training via the loss function. The results indicate that it achieves state-of-the-art performance on two benchmark datasets.

Pros:
Introduces a novel method for IDRR, leveraging logical semantics enhancement.
Achieves state-of-the-art performance on benchmark datasets.
Comprehensive experimental design, including diverse analyses.
Potential applicability to a wide range of real-world scenarios.
The paper is well-structured, making it accessible for readers (although one reviewer disagrees).
Cons:
Lacks comprehensive comparison with a broader range of state-of-the-art methods.
Issues with reproducibility due to lack of sufficient details.
Does not address potential ethical implications adequately.
Language and presentation concerns as pointed out by one reviewer.
Some aspects of the method might seem more incremental than novel. (which I agree with)